# Exploring Counterfactual Alignment Loss towards Human-centered AI

## Abstract

Deep neural networks have demonstrated impressive accuracy in supervised learning tasks. However, their lack of transparency makes it hard for humans to trust their results, especially in safe-critic domains such as healthcare. To address this issue, recent explanation-guided learning approaches proposed to align the gradient-based attention map to image regions annotated by human experts, thereby obtaining an intrinsically human-centered model. However, the attention map these methods are based on may fail to *causally* attribute the model predictions, thus compromising their validity for alignment. To address this issue, we propose a novel human-centered framework based on counterfactual generation. In particular, we utilize the counterfactual generation's ability for causal attribution to introduce a novel loss called the **C**ounter**F**actual **Align**ment (**CF-Align**) loss. This loss guarantees that the features attributed by the counterfactual generation for the classifier align with the human annotations. To optimize the proposed loss that entails a counterfactual generation with an implicit function form, we leverage the implicit function theorem for backpropagation. Our method is architecture-agnostic and, therefore can be applied to any neural network. We demonstrate the effectiveness of our method on a lung cancer diagnosis dataset, showcasing faithful alignment to humans.

## 1 Introduction

Current deep learning (DL) models, though have reached remarkable performance, are often biased towards human non-explainable features in the decision-making process Najafabadi et al. (2015); Geirhos et al. (2018). Such biases can raise concerns about trustworthiness, impeding these models from being deployed in safety-critic domains such as healthcare. To address this issue, apart from prediction accuracy, it is desirable for DL models to improve their alignment with human decision-making.

**Example 1.1.** *Take the diagnosis of lung cancer as an example. The standard protocol followed by radiologists is first to identify the region of interest (i.e., nodule), analyze its attributes (e.g., size, margin), and then assess its risks of developing into cancers Gould et al. (2013). Nevertheless, this protocol is not adhered to DL models, which primarily rely on data-driven approaches to learn label-related, but unnecessarily explainable features. As shown in Fig. 1, deep neural networks mainly utilize background features, such as those of the rib and spine, for prediction. Though these nodule-irrelevant features may endow networks with beyond human performance, they are not understandable for radiologists, thus raising severe safety concerns.*

To address this issue, many studies attempted to learn human explainable features for prediction. Examples include Liu et al. (2021b); Wang et al. (2022) that incorporated human knowledge with specialized neural network architectures; and Zheng et al. (2017); Sreedevi et al. (2022) that proposed decision-making systems based on the human cognitive model. In particular, explanation-guided learning approaches Ross et al. (2017); Ismail et al. (2021); Gao et al. (2022); Fei (2022) proposed to align the attention map of DL models to image regions annotated by human experts, thereby enforcing an intrinsically human-centered model. However, the attention map these methods are based on is obtained in a data-driven manner and not necessarily provides a causal attribution for the model's decision, which largely compromises their validity for alignment.

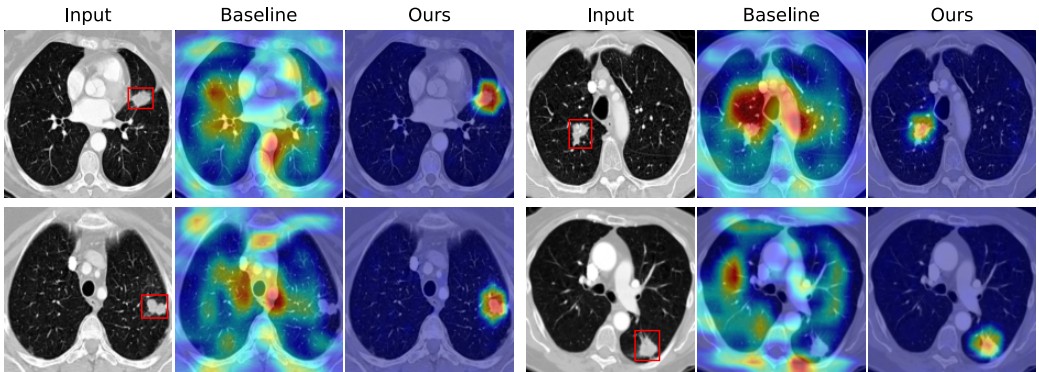

Figure 1: Saliency maps of a baseline DL model Ronneberger et al. (2015) and our method in lung cancer prediction. Lung nodules are marked with red bounding boxes.

In this paper, we propose a novel human-centered framework based on counterfactual generation. Specifically, we first leverage the counterfactual generation to identify features that *causally* attribute to the model's prediction. Then, to align these features with human annotations, we propose a novel **C**ounter**F**actual **Align**ment (**CF-Align**) loss, which punishes the network once the identified causal factors do not align with human annotations. To optimize the proposed loss that involves counterfactual generation utilizing implicit function forms, we propose an implicit gradient solver rooted in the implicit function theorem. Returning to the lung cancer example, Fig. 1 shows that our method can learn explainable features; as a contrast, features utilized by the baseline DL model are difficult to interpret.

**Contributions.** To summarize, our contributions are:

1. We propose a counterfactual alignment loss for human-centered AI.
2. We propose to leverage the implicit function theorem for optimization.
3. We achieve more accurate and explainable results than existing methods on real-world data.

## 2 RELATED WORKS

Existing works on explainable AI (XAI) can be classified in various ways Speith (2022). One such classification distinguishes between **human-centered and technology-centered** approaches, with the former focusing on the needs of specific end-users and the latter on those of AI professionals. Another classification is based on whether the method is **ante-hoc or post-hoc** explainable, which respectively refers to instinct explainability and explanation after the model has been trained.

Our method belongs to human-centered, ante-hoc XAI. Therefore, we will provide a detailed discussion of these two branches of works below, leaving the comprehensive review of other XAI methods to Arrieta et al. (2020) and Ali et al. (2023).

**Human-centered XAI** aims to develop AI models that behave in a human-understandable manner. To achieve this goal, existing methods have proposed incorporating various human priors, such as knowledge Liu et al. (2021b); Wang et al. (2022), memory Fu et al. (2014), and cognitive models (Zheng et al. (2017); Sreedevi et al. (2022), into the design of AI models. By doing so, their models could behave in a certain way that is similar to humans, thereby gaining their trust. However, most of these methods require building model-specific architectures, which makes them difficult to adapt to different settings. **In contrast**, we propose a model-agnostic loss for human-centered learning, which can be easily adapted to various task settings and neural network architectures.

**Ante-hoc XAI** refers to the design of AI models that are transparent and intrinsically explainable. Examples of such models include white-box models, such as linear regression and decision trees; hybrid models that combine a white-box model with a black-box model for improved performance Nauta et al. (2021); Wang & Lin (2021); joint prediction-explanation models that were trained to

provide both predictions and explanations Hind et al. (2019); Rieger et al. (2020); and methods that achieved explanation through architecture adjustment Zhang et al. (2018); Chen et al. (2019).

Of particular relevance to our work are **explanation-guided learning approaches** Ross et al. (2017); Ismail et al. (2021); Gao et al. (2022); Fei (2022), which proposed to align model attention or gradient with human-annotated areas. Nevertheless, these attention regions might not align with the causal factors influencing the decision-making process, potentially undermining their appropriateness for alignment purposes. **In contrast**, our method is based on the counterfactual generation that intrinsically corresponds to causal attributions, which thus ensures the alignment of the decision process with that of domain experts.

**Counterfactual explanation.** Our work is closely related to the counterfactual explanation methods Verma et al. (2020); Balasubramanian et al. (2020); Lang et al. (2021), which aimed at answering what could the outcome have changed to had input to a model had been changed in a particular way. To implement, they proposed to minimize alterations that changed the prediction. Such a modified region can be taken as the causal factor to determine the model's prediction Parafita & Vitrià (2019). Compared to the attention-based methods, this method can causally attribute the model's decision Parafita & Vitrià (2019). However, it is important to note that existing counterfactual explanation methods were designed to explain trained black-box models, while our method aims to design an intrinsically explainable model (*i.e.*, ante-hoc explainability).

## 3 PRELIMINARY

In this section, we introduce the problem setting, followed by an overview of the counterfactual explanation that our method builds upon.

**Problem setup & notations.** We consider the classification scenario, where the system includes an image $\boldsymbol{x} \in \mathcal{X} \in \mathbb{R}^p$ and a label $y \in \mathcal{Y}$ from an expert annotator. In addition to $y$, we assume the annotator also provides an explanation $\boldsymbol{e}$ that explains his/her decision for each image.

In practice, the explanation mainly refers to the region of central interest. For example, in a radiology report, the radiologist often explains his/her decision by annotating disease-related/lesion areas. Motivated by this, we assume that for each sample $(\boldsymbol{x}, y)$, the explanation can be formed as the mask $\boldsymbol{r} \in [0, 1]^p$, where $r_i = 1$ meaning that the feature $i$ belongs to the region of interest. In this regard, the training data we collect can be denoted as $\{\boldsymbol{x}_i, y_i, \boldsymbol{r}_i\}_{i=1}^n$. With this data, our objective is to learn a classification model $f_{\boldsymbol{\theta}} : \mathcal{X} \mapsto \mathcal{Y}$ that **i)** predicts $y$ accurately, and **ii)** makes its prediction based on the region $\boldsymbol{r}$.

### 3.1 COUNTERFACTUAL EXPLANATION

We first provide an overview of the counterfactual explanation method Verma et al. (2020) that our method is based on. By generating a modified image and observing its difference from the original image, this framework intends to answer the following counterfactual question: what could have the model made a different prediction had the input $X$ changed in a particular way?

For this purpose, for the classifier $f_{\boldsymbol{\theta}}$ and each sample $(\boldsymbol{x}, y)$, this framework generates a *counterfactual image* $\boldsymbol{x}^*$ with respect to the counterfactual class $y^* \neq y$:

$$\boldsymbol{x}^* := \arg\min_{\boldsymbol{x}'} \text{CE}(f_{\boldsymbol{\theta}}(\boldsymbol{x}'), y^*) + \lambda d(\boldsymbol{x}, \boldsymbol{x}'), \tag{1}$$

where CE is the cross-entropy loss, $d(\cdot)$ is a distance measure that constrains the modification to be sparse Verma et al. (2020), and $\lambda$ is regularization hyperparameter.

With such a modified image $\boldsymbol{x}^*$, we can then say the modified region $\mathbb{1}(|\boldsymbol{x}^* - \boldsymbol{x}|)$ to be the explanation of $f_{\boldsymbol{\theta}}$'s prediction, in the sense that if the pixels in this region had taken values from $\boldsymbol{x}^*$, then the prediction would have been $y^*$. Indeed, this means for a human-centered model, the counterfactual modification should belong to the annotated region of interest. This motivates our *counterfactual alignment loss*, as introduced in the next section.

**Counterfactual Explanation (CE) vs Adversarial Attack (AA).** These two methods are similar in the optimization form Szegedy et al. (2013); Wachter et al. (2017), but are intrinsically different in terms of objectives and distance measures Freiesleben (2022). Specifically, the CE hopes to identify

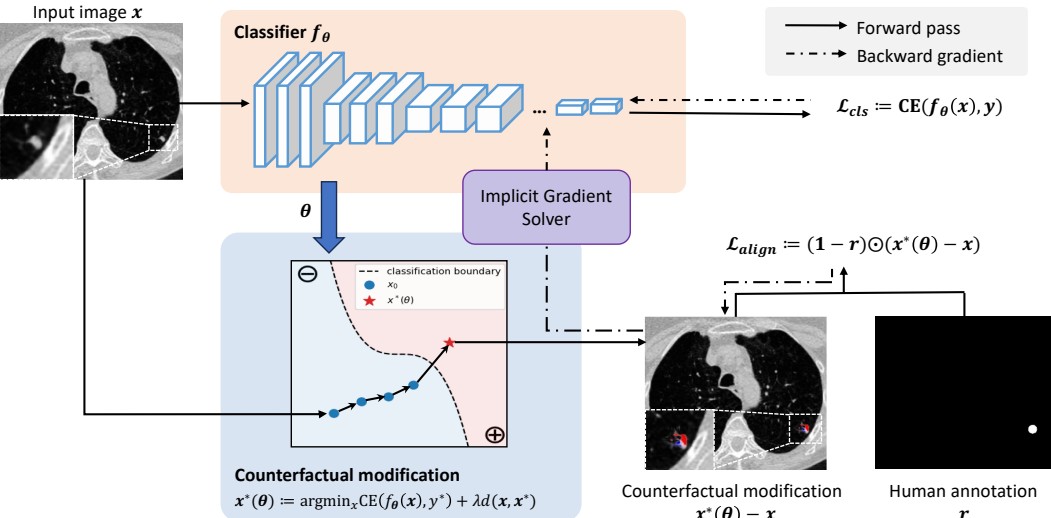

Figure 2: Overview of our method. The overall objective consists of two parts: the cross-entropy loss $\mathcal{L}_{cls}$ and the CF-Align loss $\mathcal{L}_{align}$. In the forward pass, the counterfactual image $\boldsymbol{x}^*(\boldsymbol{\theta})$ is used to compute the $\mathcal{L}_{align}$, where $\boldsymbol{x}^*(\boldsymbol{\theta})$ takes $\boldsymbol{x}$ and $\boldsymbol{\theta}$ as inputs. In the backward pass, we compute the gradient of $\mathcal{L}_{align}$ w.r.t. $\theta$ with an implicit gradient solver, then optimize the $\boldsymbol{\theta}$ such that the counterfactual modification is within the human annotations boundary $\boldsymbol{r}$.

only decision-relevant features. Therefore, the $d(\boldsymbol{x}, \boldsymbol{x}^*)$ should take into account the sparsity, which can be achieved by $\ell_0$ or $\ell_1$ norm. On the other hand, the AA hopes to generate samples whose modifications are imperceptible, for which one typically adopts $\ell_2$ or $\ell_\infty$ norm.

## 4 METHOD

In this section, we propose to align the decision basis of the model with regions provided by human experts. To this end, we propose a regularity term dubbed as the **C**ounter**F**actual **Align**ment loss (**CF-Align** loss). The goal of this loss is to enforce the network's decision basis (identified with Eq. (1)) to align with the region of interest $\boldsymbol{r}$. In this regard, the classifier will be forced to make predictions based on human-explainable features. The overall pipeline of our method is illustrated in Fig. 2.

Formally speaking, for a prediction model $f_{\boldsymbol{\theta}}$, we define the CF-Align loss as:

$$\mathcal{L}_{align}(\boldsymbol{\theta}) := \frac{1}{n} \sum_{i=1}^{n} \|(1 - \boldsymbol{r}_i) \odot (\boldsymbol{x}^*(\boldsymbol{\theta}) - \boldsymbol{x})\|_1, \tag{2}$$

where $\boldsymbol{x}^*(\boldsymbol{\theta})$ obtained from Eq. 1 is a function of the model parameter $\boldsymbol{\theta}$, $\boldsymbol{r}_i$ is a binary mask that represents the region of interest, and $\odot$ denotes the element-wise Hadamard product between matrices.

By combining with this loss to the cross-entropy loss $\mathcal{L}_{cls}(\boldsymbol{\theta})$, the overall training objective for the prediction model $f_{\boldsymbol{\theta}}$ is:

$$\mathcal{L}(\boldsymbol{\theta}) := \mathcal{L}_{cls}(\boldsymbol{\theta}) + \alpha \mathcal{L}_{align}(\boldsymbol{\theta}), \tag{3}$$

where the hyper-parameter $\alpha$ is the trade-off between classification accuracy and alignment to experts' annotation boundary. A higher value of $\alpha$ could potentially lead to improved alignment, but it might also compromise prediction accuracy because the model could leverage other features that may not be understandable to domain knowledge for making predictions. Specifically, it was well established that deep learning methods mainly focused on textual features Geirhos et al. (2018). Besides, existing works in medical imaging have also shown that microscopic features (*e.g.*, contour, curvature) Wang et al. (2021) or contextual features Liu et al. (2021a) can improve the prediction power. Moreover, apart from the lack of explainability, such features may fall outside the

realm of causal relationships grasped by domain experts and may pertain to spurious correlations. These correlations could potentially lead to non-robustness when faced with perturbations or out-of-distribution scenarios.

**Optimization.** To optimize Eq. (3), we need to compute the gradient $\nabla_{\boldsymbol{\theta}} \mathcal{L}_a(\boldsymbol{\theta})$, which involves the term $\nabla_{\boldsymbol{\theta}} \boldsymbol{x}^*(\boldsymbol{\theta})$. The main challenge lies in that the functional form of $\boldsymbol{x}^*(\boldsymbol{\theta})$ is not explicit, making it hard to derive the gradient using chain rules.

To compute this implicit gradient, we employ the Implicit Function Theorem (IFT). The idea is to note the fact that $\nabla_{\boldsymbol{x}} \arg\min_{\boldsymbol{x}} g(\boldsymbol{x}, \boldsymbol{\theta}) \equiv \mathbf{0}$ for any differential function $g$, which allows us to write $\nabla_{\boldsymbol{\theta}} \boldsymbol{x}^*(\boldsymbol{\theta})$ as the product of the inverse Hessian $H_g[\boldsymbol{x}]^{-1}$ and the mixed derivative $\nabla_{\boldsymbol{\theta}}(\nabla_{\boldsymbol{x}} g)$. Formally, we have:

**Theorem 4.1** (Inverse Function Theorem (IFT)). *Consider two vectors $\boldsymbol{x} \in \mathbb{R}^p, \boldsymbol{\theta} \in \mathbb{R}^d$, and a function $g(\boldsymbol{x}, \boldsymbol{\theta}) : \mathbb{R}^p \times \mathbb{R}^d \mapsto \mathbb{R}$. Let $\boldsymbol{x}^*(\boldsymbol{\theta}) := \arg\min_{\boldsymbol{x}} g(\boldsymbol{x}, \boldsymbol{\theta})$. In addition, suppose that the following conditions hold: i) $g$ is differentiable, ii) the $\arg\min$ is unique for each $\boldsymbol{\theta}$, and iii) the Hessian matrix $H_g[\boldsymbol{x}]$ is invertible. Then, we have:*

$$\nabla_{\boldsymbol{\theta}} \boldsymbol{x}^*(\boldsymbol{\theta}) = -H_g[\boldsymbol{x}]^{-1} \nabla_{\boldsymbol{\theta}}(\nabla_{\boldsymbol{x}} g)|_{\boldsymbol{x}^*(\boldsymbol{\theta}), \boldsymbol{\theta}}.$$

**Remark 4.2.** *For modern neural networks, the inverse Hessian is generally intractable to compute. For this case, to acquire $\nabla_{\boldsymbol{\theta}} \boldsymbol{x}^*(\boldsymbol{\theta})$, we can use linear system solvers such as the conjugate gradient solver Hestenes et al. (1952) to solve the linear equation $H_g[\boldsymbol{x}] \nabla_{\boldsymbol{\theta}} \boldsymbol{x}^*(\boldsymbol{\theta}) + \nabla_{\boldsymbol{\theta}}(\nabla_{\boldsymbol{x}} g)|_{\boldsymbol{x}^*(\boldsymbol{\theta}), \boldsymbol{\theta}} = \mathbf{0}$.*

By setting $g(\boldsymbol{x}, \boldsymbol{\theta}) := \mathrm{CE}(f_{\boldsymbol{\theta}}(\boldsymbol{x}), y^*)$, we can compute the implicit gradient $\nabla_{\boldsymbol{\theta}} \boldsymbol{x}^*(\boldsymbol{\theta})$ and thereby the gradient $\nabla_{\boldsymbol{\theta}} \mathcal{L}_a$ with the chain rule. Besides, it is important to note that our proposed CF-align loss is model-agnostic. Therefore, it can be easily plugged into the training of various neural networks for human-centered learning.

**Extension to attributes $\boldsymbol{a}$.** In addition to the class label $y$, Eq. 2 also applies when the annotated regions $\boldsymbol{r}$ explains its decision to attributes annotations $\boldsymbol{a}$, which will subsequently affect the label $y$. For instance, in the context of lung cancer diagnosis, $\boldsymbol{a}$ pertains to clinical attributes related to nodules, which include factors such as size, margin, spiculation, and more. During the diagnostic process, clinicians commonly annotate the nodule's region and provide assessments of these attributes, which serve as the basis for their diagnostic decisions.

If these attributes are provided during training, then we can leverage them into our framework. In this case, the overall classifier can be written as $g_{\boldsymbol{\gamma}} \circ f_{\boldsymbol{\theta}}$, where $f_{\boldsymbol{\theta}} : \mathcal{X} \mapsto \mathcal{A}$ and $g_{\boldsymbol{\gamma}} : \mathcal{A} \mapsto \mathcal{Y}$. To determine which attributes to modify, we first employ Zhao et al. (2023) to identify key attributes $\bar{\boldsymbol{a}}$ that have causal influences on $y$, and then generate the counterfactual image by:

$$\boldsymbol{x}^*(\theta) := \arg\min_{\boldsymbol{x}'} \mathrm{CE}(f_{\boldsymbol{\theta}}(\boldsymbol{x}'), \bar{\boldsymbol{a}}^*) + \lambda d(\boldsymbol{x}, \boldsymbol{x}'). \tag{4}$$

To train $(\boldsymbol{\theta}, \boldsymbol{\gamma})$, we can optimize the objective function Eq. 3 such that the CE loss is replaced with $\mathcal{L}_{cls}(\boldsymbol{\theta}, \boldsymbol{\gamma})$ to account for the classifier $g_{\boldsymbol{\gamma}}$ and that the $\boldsymbol{x}^*(\boldsymbol{\theta})$ is generated through Eq. 4. For a new sample $\boldsymbol{x}$ to predict, we first use $f_{\boldsymbol{\theta}(\boldsymbol{x})}$ to predict attributes $\boldsymbol{a}$ based on explainable features, then predict $y$ based on attributes $\boldsymbol{a}$.

## 5 EXPERIMENT

To demonstrate the practicability of enhancing the explainability within our method, we apply our model to the pulmonary nodule benign/malignant classification. It's important to clarify that our objective is not to attain state-of-the-art performance on this particular task but rather to leverage it as an illustrative example to emphasize our key point.

### 5.1 DATASET & IMPLEMENTATION

We consider the LIDC-IDRI dataset Armato III et al. (2011), which contains imaging data obtained from clinical thoracic CT scans, as well as fine-grained annotations (nodule bounding boxes, malignancy scores, and attributes information) provided by experienced physicians. Prior to analysis, we

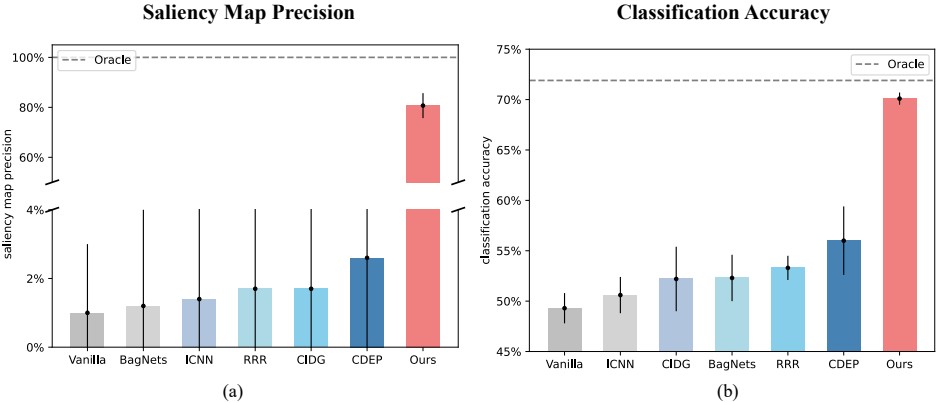

Figure 3: Comparison with baselines on the LIDC-IDRI dataset. We use (a) saliency map precision to evaluate model explainability and (b) accuracy to evaluate classification performance.

preprocess the images by resampling the pixel space to a resolution of $1 \times 1 \times 1 \text{mm}^3$ and normalizing the intensity based on a window center of $\text{HU} = -600$ and a window width of $\text{HU} = 1600$. In order to ascertain whether our method can learn the same features as those used by physicians, we opt to take the entire 2D slice as input, without explicitly cropping out the nodules. Regarding the classification labels, each sample was scored by physicians, with 3 or lower meaning benign ($y = 0$), while those with a score of 4 or higher are considered malignant ($y = 1$). Besides malignancy labels, each sample is accompanied by six clinical attributes: subtlety, calcification, margin, spiculation, lobulation, and texture. A comprehensive description of these attributes and their influence on the disease label can be found in Appx. B.1. In addition, physicians also provide annotations for the nodule bounding box for each sample, which serves as the foundation for annotating attributes and determining the disease label. We divide the dataset into three subsets: training ($n = 731$ nodules), validation ($n = 244$ nodules), and test ($n = 238$ nodules). To avoid possible label leaking, this partition is based on individual patients.

We implement a seven-layer convolutional neural network (Fig. 7) to parameterize $f_{\boldsymbol{\theta}}$. We use the Adam optimizer to train our model, with the learning rate set to $0.001$, batch size set to $128$, and epochs set to $300$. During the training, we iteratively optimize over the classification loss $\mathcal{L}_{cls}$ and the alignment loss $\mathcal{L}_{align}$. To generate the counterfactual image, we adopt the LatentCF method Balasubramanian et al. (2020), which first encodes the image into the latent space and then modifies the latent code for generation. To estimate the implicit gradient $\nabla_{\boldsymbol{\theta}} \mathcal{L}_{align}$, we use the conjugate gradient method implemented in the TorchOpt package Ren et al. (2022). We use Grad-Cam Selvaraju et al. (2017) to compute the saliency map for each method. For implementation of compared baselines, we directly load their published codes. To remove the effect of randomness, we repeat all experiments using three different seeds.

## 5.2 COMPARISON WITH BASELINES

We evaluate the classification accuracy and explainability of our method and baselines.

**Compared baselines.** Firstly, we compare with several ante-hoc explainable AI baselines, namely **i) BagNets** Brendel & Bethge (2019) that integrated the white-box bag-of-features model with a deep neural network to achieve both explainability and performance; **ii) CDEP** Rieger et al. (2020) that required the model to produce a prediction as well as an explanation (*i.e.*, multi-tasks learning); **iii) ICNN** Zhang et al. (2018) that modified the architecture of the neural network to achieve object-centered explainability; **iv) RRR** Ross et al. (2017) and **v) CIDG** Chang et al. (2021) that constrains the gradient of the input image to be aligned with annotations from radiologists.

Besides, to achieve a comprehensive comparison, we also include the **vi) Vanilla** method that optimizes only the classification loss and the **vii) Oracle** method that takes only features annotated by the physicians (*i.e.*, the cropped-out nodule regions) as input.

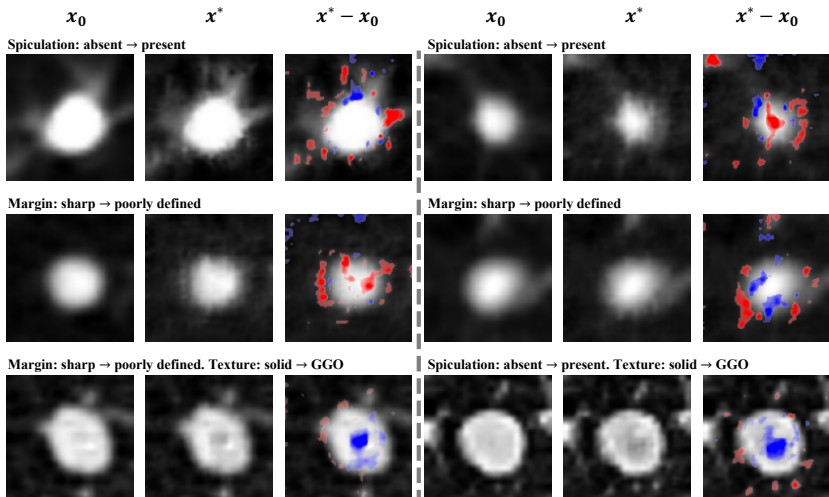

Figure 4: Visualization of the counterfactual images of six examples. For each example, the left column ($x_0$), the middle column ($x^*$), and the right column ($x^* - x_0$) respectively represent the original image, the counterfactual image, and the modified regions. Regions with positive modifications are highlighted in red, while those with negative modifications are marked in blue. The corresponding attribute changes are noted in the title.

**Metrics.** To evaluate model explainability, we employ the saliency map precision, which is defined as the ratio of the saliency map area within the nodule bounding box to the total area of the saliency map. To evaluate the classification performance, we report the accuracy metric.

**Comparison over explainability.** We report the saliency map precision of our method and baselines in Fig. 3 (a). Firstly, we can observe that our method reaches a high saliency precision of $81\%$, which means that the decision basis of our model is well aligned with that of the physicians. Such a promising result can be attributed to the fact that our counterfactual generation method can accurately identify the causes for model decision, and that Thm. A.1 provides an efficient way to estimate the implicit gradient and makes the alignment loss easy to optimize.

Furthermore, we have also noticed that other baseline methods struggle to discern features within nodule bounding boxes when making predictions. This implies that their diagnostic decisions lack explainability for physicians. To comprehend such a result, one should note that BagNets Brendel & Bethge (2019), ICNN Zhang et al. (2018), and CDEP Rieger et al. (2020) proposed no explicit constraint to learn explainable features. Therefore, for input with complicated backgrounds such as the thoracic CT, these methods can be easily biased by the contextual features in the image. Meanwhile, though methods such as RRR Ross et al. (2017) and CIDG Chang et al. (2021) explicitly constrained the gradient of the model to human decision areas, the gradient method itself has been found unable to fully represent model decision boundary Jain & Wallace (2019); Grimsley et al. (2020), making their identified areas failed to align with ground-truth annotations.

**Comparison over classification.** We report the classification accuracy in Fig. 3 (b). As shown, our method significantly outperforms the baselines. This outcome is a product of our method's ability to harness expert-explainable features that can be consistently applied to test data. In contrast, the baselines may utilize other contextual features for decisions, which are mainly pseudo-correlation and are hard to transfer.

Besides, it is also worth noting that in our results, even the oracle method can only achieve a classification accuracy of $72\%$. In contrast, some work Shen et al. (2017); Xie et al. (2018); Wu et al. (2018) on pulmonary nodule classification claimed a classification accuracy over $99\%$. This gap can be mainly explained by the Shen et al. (2017); Xie et al. (2018); Wu et al. (2018) elimination of uncertain or challenging samples (those with a score of 3 in the data) from the test dataset in Shen et al. (2017); Xie et al. (2018); Wu et al. (2018). Once again, we would like to emphasize that our experiment is not primarily aimed at surpassing existing state-of-the-art classification methods.

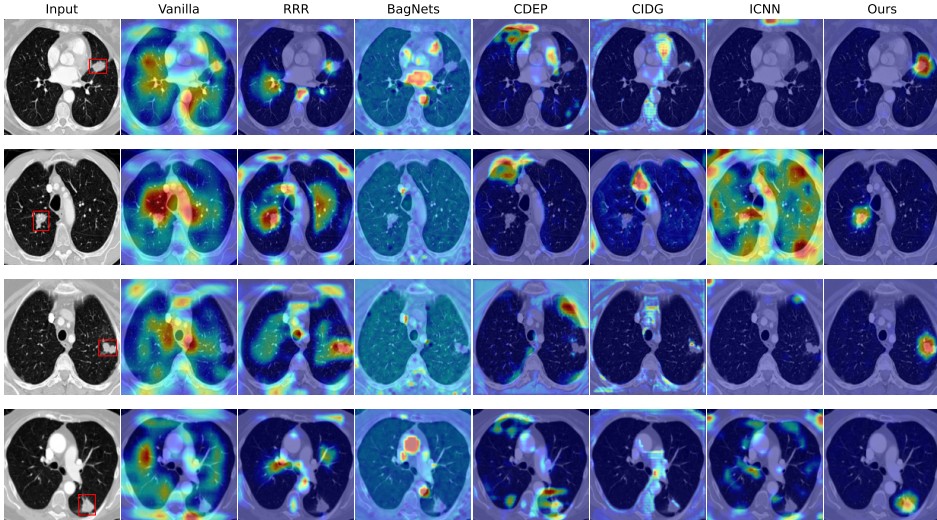

Figure 5: Saliency map visualization of our method and compared baselines. Nodules are marked by red bounding boxes. More examples can be found in Fig. 8.

Instead, our primary goal is to showcase the model's explainability and its potential benefits for classification tasks.

**Visualization of counterfactual images.** The counterfactual images generated are presented in Fig. 4. For each nodule, the left, middle, and right columns correspond to the original image, the counterfactual image, and the modified regions, respectively. As we can see, the counterfactual modifications generally correspond to one or more attributional features. For instance, the two nodules in the first row were transformed from benign to malignant by introducing spiculation around the nodule. Similarly, the two benign nodules in the 2nd row are classified to be malignant after modifying the original sharp margin to poorly defined ones[1]. These results demonstrate that our counterfactual generation process can effectively identify fine-grained visual features, thereby enabling the diagnosis decision of our model to align more closely with those made by humans.

### 5.3 VISUALIZATION OF SALIENCY MAP

To further evaluate the explainability of our method and the ability of counterfactual generation to localize fine-grained nodule features, we visualize the saliency maps (Fig. 5) and the generated counterfactual images (Fig. 4).

**Visualization of saliency maps.** The saliency maps of our method and compared baselines are shown in Fig. 5. As we can see, our method can accurately identify nodule-related features, while baseline methods mostly focus on areas irrelevant to the nodule, such as the rib, pleural, and spine. These results verify that our method can effectively constrain the neural network to learn human-centered features, making our diagnosis process intrinsically explainable.

### 5.4 RADIOLOGY REPORT GENERATION

To present our explanations to human end-users in a clear and accessible manner, we use a Large Language Model (LLM)-based interface to generate a structured report. Specifically, we employ GPT-4 as a backend through prompting Brown et al. (2020). The prompt we provide to the language model contains three parts, namely a task description, the predictions, and an explanation that is composed of the disease label, attributes, and saliency map from our classifier.

To ensure the report is well-organized and maintains a professional appearance, we present GPT-4 with various standard examples provided by radiologists from Irvin et al. (2019) and instruct it to

---

[1]Please see `https://pylidc.github.io/annotation.html`

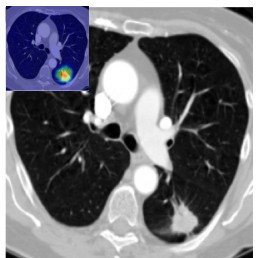

> **PATIENT ID**: LIDC-IDRI-0001
> **TECHINIQUE**: CT scan of the lung was done using 2.5 x 0.7 mm slices.
> **FINDINGS**:
> There is a poorly defined 3cm solid, non-calcified nodule with spiculation and lobulation in the lower-right lung.
> **IMPRESSIONS**:
> 3cm solid, non-calcified nodule found in the lower-right lung. The features may suggest a malignant etiology. Further characterization is recommended with a CT scan and follow-up at 3-6 months.

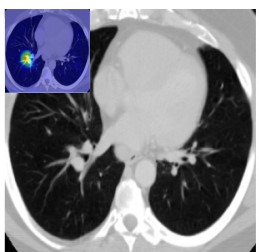

> **PATIENT ID**: LIDC-IDRI-0961
> **TECHINIQUE**: CT scan of the lung was done using 3 x 0.7 mm slices.
> **FINDINGS**:
> There is a 8mm nodule in the middle-left lung that is solid and non-calcified with a sharp margin.
> **IMPRESSIONS**:
> 8mm solid, non-calcified nodule found in the middle-left lung. The features suggest the nodule is stable and highly likely benign. In the absence of prior scans, it's recommended to schedule a CT at 6-12 months for confirmation, followed by subsequent CTs at 3 and 5 years.

Figure 6: Radiology reports generated by GPT-4 that summarize our explanations composed of the saliency map (attached to the original image in the left corner, findings regarding attributes, and the final prediction and suggestions.

generate a report that adheres to the format of these examples. Specifically, the prompt we provide to GPT-4 is formated as follows:

1. **Task description:** You are a machine that explains the decision of a deep learning model to human end-users.

2. **Predictions and explanations:** [size][attributes] lesion found in [position]. (if malignant) These features may suggest a malignant etiology. (if benign) These features suggest the nodule is stable and highly likely benign. [follow-up schedule suggestions].

3. **Canonical examples:** [reports by physicians][2].

To illustrate the fidelity of the generated reports, we provide two examples in Fig. 6. In the 1st example, the nodule exhibits malignant features such as poorly defined margin, spiculation, and lobulation, while in the 2nd example, the nodule displays benign features such as solid texture and sharp margin. Our evaluation focuses on the accuracy and comprehensiveness of the generated reports. As shown, the findings section of the report clearly states the position and attributes of the nodules. Based on such findings, the impressions section provides a correct diagnosis and a detailed explanation of the underlying reasons for such a diagnosis. Further, physicians offer suggestions on follow-up examination schedules based on the disease severity. The generated reports summarize these explanations, which can effectively communicate the diagnosis results to human readers.

## 6 CONCLUSIONS

In this paper, we present a counterfactual-based framework aimed at aligning the model with human-explainable features for prediction. To this end, a counterfactual alignment loss is introduced to ensure that the model only modifies regions within human annotations during counterfactual generation. To optimize this loss, we leverage the implicit function theorem to compute the gradient of the alignment loss, which involves implicit forms in the counterfactual generation process. Our framework's effectiveness is demonstrated by its ability to guide the prediction model in leveraging nodule-related features for the diagnosis of pulmonary nodule malignancy.

---

[2]Please refer to Appx. A.3 for the examples we use.

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

# APPENDIX

## A METHOD

### A.1 PRELIMINARY ON MATRIX DERIVATIVE

For $y = f(\boldsymbol{x})$, where $f : \mathbb{R}^p \mapsto \mathbb{R}$ and $x \in \mathbb{R}^p$, the **gradient vector** is defined as:

$$\nabla_x f := \left[ \frac{\partial f}{\partial x_1}, \cdots, \frac{\partial f}{\partial x_p} \right]^T.$$

For $\boldsymbol{x} = g(\boldsymbol{\theta})$, where $g : \mathbb{R}^m \mapsto \mathbb{R}^p$, $\boldsymbol{\theta} \in \mathbb{R}^m$, and $\boldsymbol{x} \in \mathbb{R}^p$, the **Jacobian matrix** is defined as:

$$\nabla_{\boldsymbol{\theta}} \boldsymbol{x} := \begin{bmatrix} \frac{\partial x_1}{\partial \theta_1} & \cdots & \frac{\partial x_1}{\partial \theta_m} \\ \vdots & \ddots & \vdots \\ \frac{\partial x_p}{\partial \theta_1} & \cdots & \frac{\partial x_p}{\partial \theta_m} \end{bmatrix}_{p \times m}.$$

For $y = f(\boldsymbol{x})$, the **Hessian matrix** is defined as the Jacobian of the gradient vector. That is:

$$H_f[\boldsymbol{x}] := \nabla_{\boldsymbol{x}}(\nabla_{\boldsymbol{x}} f) = \begin{bmatrix} \frac{\partial^2 f}{\partial x_1^2} & \cdots & \frac{\partial^2 f}{\partial x_1 \partial x_p} \\ \vdots & \ddots & \vdots \\ \frac{\partial^2 f}{\partial x_p \partial x_1} & \cdots & \frac{\partial^2 f}{\partial x_p^2} \end{bmatrix}_{p \times p}$$

### A.2 PROOF OF THE INVERSE FUNCTION THEOREM (IFT)

**Theorem A.1** (IFT). *Consider two vectors $\boldsymbol{x} \in \mathbb{R}^p, \boldsymbol{\theta} \in \mathbb{R}^m$, and a function $f(\boldsymbol{x}, \boldsymbol{\theta}) : \mathbb{R}^p \times \mathbb{R}^m \mapsto \mathbb{R}$. Let $\boldsymbol{x}^*(\boldsymbol{\theta}) := \arg\min_{\boldsymbol{x}} f(\boldsymbol{x}, \boldsymbol{\theta})$. Suppose that the following regularities hold:*

1. *$f$ is continuous and differentiable,*

2. *the $\arg\min$ is unique for each $\boldsymbol{\theta}$,*

3. *the Hessian matrix $H_f[x]$ is invertible.*

*Then, we have:*

$$\nabla_{\boldsymbol{\theta}} \boldsymbol{x}^*(\boldsymbol{\theta}) = -H_f[\boldsymbol{x}]^{-1} \cdot \nabla_{\boldsymbol{\theta}}(\nabla_{\boldsymbol{x}} f)|_{\boldsymbol{x}^*(\boldsymbol{\theta}), \boldsymbol{\theta}}.$$

*Proof.* Since the $\boldsymbol{x}^*(\boldsymbol{\theta})$ is defined by $\arg\min_{\boldsymbol{x}} f(\boldsymbol{x}, \boldsymbol{\theta})$, we have:

$$\nabla_{\boldsymbol{x}} f|_{\boldsymbol{x}^*(\boldsymbol{\theta}), \boldsymbol{\theta}} = [\mathbf{0}]_{p \times 1}.$$

Then, we have:

$$\nabla_{\boldsymbol{\theta}}(\nabla_{\boldsymbol{x}} f|_{\boldsymbol{x}^*(\boldsymbol{\theta}), \boldsymbol{\theta}}) = \nabla_{\boldsymbol{\theta}}([\mathbf{0}]_{p \times 1}) = [\mathbf{0}]_{p \times m}.$$

Now, applying the law of total derivation, we have:

$$\nabla_{\boldsymbol{\theta}}(\nabla_{\boldsymbol{x}} f|_{\boldsymbol{x}^*(\boldsymbol{\theta}), \boldsymbol{\theta}}) = \{ \nabla_{\boldsymbol{x}}(\nabla_{\boldsymbol{x}} f) \cdot \nabla_{\boldsymbol{\theta}} \boldsymbol{x}^* + \nabla_{\boldsymbol{\theta}}(\nabla_{\boldsymbol{x}} f) \}|_{\boldsymbol{x}^*(\boldsymbol{\theta}), \boldsymbol{\theta}}$$

Denote $H_f[\boldsymbol{x}] := \nabla_{\boldsymbol{x}}(\nabla_{\boldsymbol{x}} f)$ as the Hessian matrix. The above two equations together mean:

$$\{ H_f[\boldsymbol{x}] \cdot \nabla_{\boldsymbol{\theta}} \boldsymbol{x}^*(\boldsymbol{\theta}) + \nabla_{\boldsymbol{\theta}}(\nabla_{\boldsymbol{x}} f) \}|_{\boldsymbol{x}^*(\boldsymbol{\theta}), \boldsymbol{\theta}} = [\mathbf{0}]_{p \times m},$$

which shows the statement in the theorem. $\square$

### A.3 PROMPT EXAMPLES

## B  EXPERIMENT

### B.1  DESCRIPTION OF NODULE ATTRIBUTES

Table 1: Attributes of pulmonary nodules

| Name | Description | How to convert to a binary label |
|------|-------------|----------------------------------|
| Subtlety | Size of the nodule | Subtle: score $\leq 4$; Obvious: score $= 5$ |
| Calcification | Whether calcification is present | Absent: score $= 6$; Present: score $\leq 5$ |
| Margin | Whether the margin is well-defined | Sharp: score $= 5$; Pooly-defined: score $\leq 4$ |
| Spiculation | Whether spiculation is present | Absent: score $= 1$; Present: score $\geq 2$ |
| Lobulation | Whether lobulation is present | Absent: score $= 1$; Present: score $\geq 2$ |
| Texture | Radiographic solidity | Solid: score $= 5$; GGO/Mixed: score $\leq 4$ |

### B.2  NETWORK ARCHITECTURE

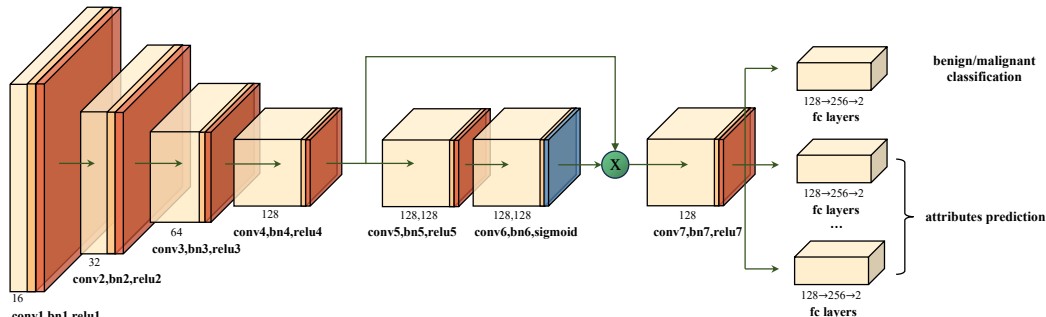

Figure 7: Architecture of the convolutional neural network used in the experiment.

## B.3 EXTRA VISUALIZATION RESULTS

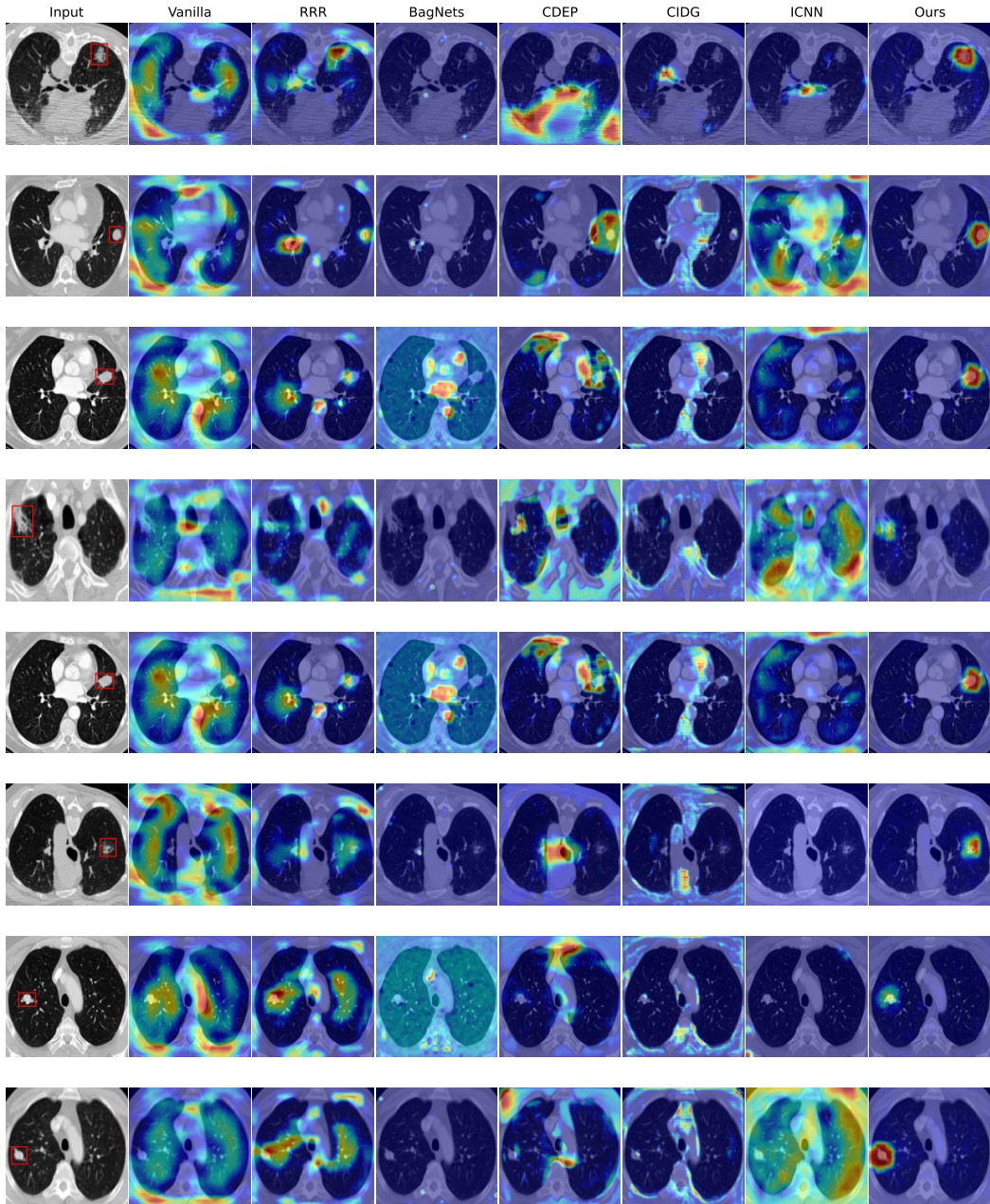

Figure 8: Visualization of the saliency maps of different methods. Each row indicates an instance.

