# OpenReview forum: "Exploring Counterfactual Alignment Loss towards Human-Centered AI"
_ICLR.cc/2024/Conference — ICLR 2024 Conference Withdrawn Submission_

### Official Review · Reviewer_orMV · 2023-10-28

**Soundness:** 3 good
**Presentation:** 4 excellent
**Contribution:** 3 good
**Rating:** 6
**Confidence:** 4

**Summary:**

The authors propose a novel explanation-guided training procedure aimed at obtaining inherently explainable models that align with human decision processes. They leverage counterfactual explanations to ensure causal alignment with human annotations, and employ the implicit function theorem due to the intractability of gradients. The experiments on a lung cancer dataset demonstrate promising results.

**Strengths:**

- The paper addresses a relevant and important topic in the field of AI, namely the training of explainable models that align with human decision-making processes.
- The level of detail with which design decisions by the authors are made is on point, which enhances the clarity of the work. I also like how differences to other approaches are highlighted, e.g., differences between counterfactual explanations and adversarial attacks (equations very similar)
- The experiments conducted are sensible and employ meaningful metrics, specifically saliency map precision and accuracy.

**Weaknesses:**

- The primary limitation of this work is the use of only one dataset in the experiment section. For this particular dataset, the Vanilla model, i.e., a model not levering human guidance, fails to accurately solve the classification task. It would be interesting to see results for datasets where Vanilla model can solve the task, but using wrong (or unintuitive) features, for example in the shortcut learning setting. This would align better with the statement the authors made in the introduction (“DL models achieve remarkable results, but with non-explainable features”)

**Questions:**

- What is the computational overhead caused by the introduction of the new loss term, specifically for the conjugate gradient solver?
- Are models trained with the alignment loss from scratch? Or is this procedure used to “correct” pre-trained models?

---

### Official Review · Reviewer_bgGJ · 2023-10-31

**Soundness:** 2 fair
**Presentation:** 2 fair
**Contribution:** 2 fair
**Rating:** 3
**Confidence:** 4

**Summary:**

This paper delves into the application of detailed human rationales, specifically bounding boxes of the input image that contain the causal attribute of the image, to enhance image classification performance. The authors introduce a counterfactual alignment loss that encourages the model to concentrate on the same regions as the provided annotations. The model's effectiveness is demonstrated using a lung cancer diagnosis dataset, showing meaningful interpretations.

**Strengths:**

1. The integration of counterfactual reasoning with human rationales is an intriguing approach. The paper's motivation is well-articulated.
2. The paper is generally well-written and easy to comprehend, although there are areas where improvements could be made.

**Weaknesses:**

1. The authors appear to lack a comprehensive understanding of the current state of the field. Numerous papers have already explored the use of human rationales to enhance prediction accuracy [1,2,3]. Even within the specific context of lung cancer prediction, prior work has demonstrated the value of using human annotations [4]. It is crucial for the authors to distinguish their method, theoretically and empirically, from previously published work.
2. The empirical evaluation of the method is somewhat lacking, with only one lung cancer prediction dataset used. There are numerous other vision tasks, such as Visual Question Answering, that could demonstrate the model's effectiveness. Furthermore, even within lung cancer prediction, there are multiple datasets available, such as NLST (https://cdas.cancer.gov/learn/nlst/images/). It is not feasible to accept this paper based solely on results from one dataset.
3. The results in the paper are presented as a bar plot without any numerical values. For a scientific paper, it is imperative to include all raw data (accuracies) in a tabular format.

Here are the related references that I found on google scholar in just 5 minutes:
[1] Qiao, Tingting, Jianfeng Dong, and Duanqing Xu. "Exploring human-like attention supervision in visual question answering." Proceedings of the AAAI Conference on Artificial Intelligence. Vol. 32. No. 1. 2018.
[2] Selvaraju, Ramprasaath R., et al. "Casting your model: Learning to localize improves self-supervised representations." Proceedings of the IEEE/CVF Conference on Computer Vision and Pattern Recognition. 2021.
[3] Selvaraju, Ramprasaath R., et al. "Casting your model: Learning to localize improves self-supervised representations." Proceedings of the IEEE/CVF Conference on Computer Vision and Pattern Recognition. 2021.
[4] Mikhael, Peter G., et al. "Sybil: A validated deep learning model to predict future lung cancer risk from a single low-dose chest computed tomography." Journal of Clinical Oncology 41.12 (2023): 2191-2200.

**Questions:**

CE typically stands for cross entropy. Please do not use it as an acronym for Counterfactual Explanation. In fact, in Figure 2, you use CE for cross entropy yourself.

---

### Official Review · Reviewer_wsVV · 2023-11-01

**Soundness:** 2 fair
**Presentation:** 2 fair
**Contribution:** 2 fair
**Rating:** 3
**Confidence:** 4

**Summary:**

In this paper, the authors present a counterfactual-based framework aimed at aligning the model with human explainable features for prediction. A counterfactual alignment loss is introduced to ensure that model only modifies regions within human annotations during counterfactual generation. Experiments demonstrate the effectiveness of the proposed method.

**Strengths:**

S1: This paper considers a real-world scenario, such as the lung cancer diagnosis dataset, which is significant important in safe-critic domain.

S2: A counterfactual-alignment loss is used to improve the performance within the human annotations.

**Weaknesses:**

W1: The method seems to be incremental and does not have any novelty.

W2: The experimental results is very poor, and many ablation studies is missing, such as the used loss of ce or alignment loss.

W3: How to get these counterfactual images?

**Questions:**

See Weaknesses